# Interrupted Lives: Welfare Considerations in Wildlife Rehabilitation

**DOI:** 10.3390/ani13111836

**Published:** 2023-06-01

**Authors:** Michelle Willette, Nicki Rosenhagen, Gail Buhl, Charles Innis, Jeff Boehm

**Affiliations:** 1The Raptor Center, St. Paul, MN 55108, USA; gailbuhl@umn.edu; 2Progressive Animal Welfare Society, Lynnwood, WA 98087, USA; nrosenhagen@paws.org; 3New England Aquarium, Boston, MA 02110, USA; cinnis@neaq.org; 4The Marine Mammal Center, Sausalito, CA 94965, USA; boehmj@tmmc.org

**Keywords:** animal welfare, wildlife rehabilitation, clinical wildlife medicine, urban wildlife, birds, sea turtles, marine mammals

## Abstract

**Simple Summary:**

Wildlife rehabilitation is the practice of caring for sick, injured, or orphaned wild animals with the goal of releasing them back to the wild. This review demonstrates the complexity of considerations rehabilitators and veterinarians face while trying to optimize the welfare of wildlife in need of care and rehabilitation. The welfare of animals, including the thousands of wild animals presented for rehabilitation each year in the United States, is of increasing concern to the public. Almost all wildlife rehabilitation in the United States requires practitioners to have a federal or state permit and is privately funded. The process of rehabilitation is inherently stressful for wildlife and maintaining the individual animal’s welfare at the center of the rehabilitation process requires deliberate, timely, and humane decision making. The welfare of wild animals can be improved by preventing human-related causes of admission, providing much-needed resources and support for those animals in rehabilitation, further developing evidence-based wildlife rehabilitation methods and welfare measures, increasing engagement of the veterinary profession, harmonizing regulatory oversight with standards of care, training, and accountability, and raising public awareness regarding the steps that can be taken to mitigate the number of wild animals in need of rehabilitation.

**Abstract:**

Each year in the United States, thousands of sick, injured, or displaced wild animals are presented to individuals or organizations who have either a federal or state permit that allows them to care for these animals with the goal of releasing them back to the wild. The purpose of this review is to demonstrate the complexity of considerations rehabilitators and veterinarians face while trying to optimize the welfare of wild animals in need of care and rehabilitation. The process of rehabilitation is inherently stressful for wildlife. Maintaining an animal’s welfare during the rehabilitation process—from initial contact and tria+ge to the animal’s euthanasia, release, or captive placement—requires deliberate, timely and humane decision making. The welfare of wild animals can be improved by preventing human-related causes of admission, providing resources and support for wildlife rehabilitation (almost all rehabilitation in the United States is privately funded and access to veterinary care is often limited); further developing evidence-based wildlife rehabilitation methods and welfare measures, attracting more veterinary professionals to the field, harmonizing regulatory oversight with standards of care, training, and accountability, and increasing public education.

## 1. Introduction

The welfare of animals, including wildlife, is of increasing concern to the public. In the United States (US), it is estimated that hundreds of thousands of sick, injured or orphaned wild animals are presented for rehabilitation annually [1] (Ed Clark, Personal Communication, 2010). Causes for admission are myriad, from displaced juveniles to disease or trauma, and most causes are considered anthropogenic or human-caused [2,3,4,5,6,7,8,9,10,11]. Wildlife rehabilitation is the practice of caring for these animals with the goal of releasing them back to the wild [12]. Approximately half of the animals admitted are birds; the remainder are mainly urban terrestrial mammals and some herptiles [3,4,13]. Rehabilitation of marine mammals and sea turtles constitute smaller, specialized segments in the rehabilitation field. Zoos and aquaria often rehabilitate wildlife and frequently house non-releasable animals.

At its core, animal welfare is a concept that people use to express their concerns about how our actions affect the quality of life of animals. This concern for the welfare of animals has expanded from its origins of utilizing animals in food production and research to incorporating almost every interaction humans have with animals including wildlife, both captive and free-ranging [5,14,15]. However, there is no universal definition of animal welfare. For this article, the authors employ the following definitions of animal welfare: how an animal is coping with the conditions in which it lives [16] and the physical and mental state of an animal in relation to the conditions in which it lives and dies [17]. 

The process of rehabilitation is inherently stressful for wildlife and has significant impacts on their welfare [18,19,20]. When discussing “stress” the authors refer to any challenge to the animal’s homeostasis from a broad range of physical, mental, or environmental factors or stressors. While short-term stress can enhance homeostasis (eustress), chronic stressors lead to distress, which can endanger health, wellbeing, and the animal’s ability to cope [21,22,23,24]. 

Wildlife rehabilitation melds applicable regulatory policy, any available standards of care, and ethics for the care of ill, injured, or displaced animals. There is a need to clarify the welfare impacts of various rehabilitation methods and management decisions [2,3,25]. The aim of this paper is to demonstrate the complexity of considerations that rehabilitators and veterinarians face while trying to optimize the welfare of wildlife in need of care and rehabilitation across multiple taxa.

## 2. Wildlife Rehabilitation

There is little doubt that humans have cared for injured and orphaned native wildlife throughout our history. Numerous early texts speak to the care of animals, including wildlife, and the recognition that humans and animals suffer from the same diseases [26,27]. In the US, formal interest in wildlife conservation and animal welfare arose concurrent with the environmental movement of the 1960s and 1970s. In 1972, the International Wildlife Rehabilitation Council (IWRC) was formed followed by the National Wildlife Rehabilitation Association (NWRA), and with them emerged a more formal field of wildlife rehabilitation [12,28,29]. 

The primary goal of wildlife rehabilitation is to benefit the individual animal. Given the extent of anthropogenic causes of wildlife harm, many people believe we have a moral obligation to help [30,31]. In reality, wildlife rehabilitation has risks and benefits to individuals, populations of animals, and, by extension, our shared environment. Benefits include preserving biodiversity, environmental monitoring, augmenting wildlife management, scientific research, advancement of wildlife studies and veterinary medicine, informing public policy and public health, and public education, outreach, and service [32,33,34,35,36]. Risks include poor animal welfare, impacts on human, domestic and wildlife health, and waste of financial and human resources [37,38,39]. 

The scope of this paper is limited to wildlife rehabilitation as practiced in the US. Due to the predominantly urban focus of US wildlife rehabilitation programs, relatively few species rehabilitated are currently endangered or threatened. However, for certain taxa (sea turtles, marine mammals) as well as in other countries, wildlife rehabilitation, translocation, and reintroduction programs may be an important component of conservation programs [40]. Compared to the US, other countries may have more of a continuum between rescue centers, rehabilitation facilities, animal sanctuaries, and zoos [41,42,43]. 

Increasingly, zoos and aquaria in the US are engaging in native wildlife rescue and rehabilitation. The Association of Zoos and Aquariums (AZA) and its member zoos and aquariums partner with regulatory authorities, and often local wildlife rehabilitators, in rescue events to improve conservation outcomes and animal welfare and engage people to care about wildlife. This provides opportunities for cross-training between zoological and rehabilitation facilities and additional support for rehabilitators. Zoos and aquariums are also frequent placement sites for non-releasable wildlife.

In the US, wildlife rehabilitation generally requires a permit, and depending on the taxa, a variety of federal, state, and local agencies have regulatory oversight. Federal agencies oversee the rehabilitation of migratory birds, marine mammals, sea turtles, and federally endangered and threatened species, while states (usually the departments of fish and game, natural resources, animal health and/or agriculture) regulate the rehabilitation of other wildlife within their borders [44,45].

The Animal Welfare Act (AWA) enacted in 1966 is the only federal law in the US that regulates the treatment of animals in research, exhibition, transport, and by dealers [46]. Currently, it only covers some mammalian and avian species. The AWA is enforced by the US Department of Agriculture (USDA), Animal Plant Health Inspection Service (APHIS), Animal Care (AC). It is separate and apart from regulatory oversight of wildlife rehabilitation, and unless rehabilitators are conducting activities with species covered by the AWA, they are not subject to its regulations. Marine mammal rehabilitation is an exception. Facilities look to the AWA for captive care requirements and any research projects would be subject to their oversight.

Individual states have taken various approaches to wildlife rehabilitation. Some states do not address the practice at all, whereas in other states, it is not permitted, and violation may result in prosecution. These states prefer to “let nature take its course”. In some states where wildlife rehabilitation is not permitted, there may be a process to place a limited number of high-profile “orphans” in zoos or other appropriate organizations. These orphans would otherwise starve to death or inappropriately interact with people. 

Regardless of the level of regulatory oversight, there are a finite number of outcomes for admitted wildlife: euthanasia or death-in-care, successful rehabilitation and release to the wild, or permanent placement in human care. Despite treatment and rehabilitation, ultimately, many wild animals cannot or should not return to the wild. These animals could be considered for captive placement into educational, foster, or breeding programs. There are regulatory as well as significant ethical considerations regarding captive placement, as there are limited numbers of suitable facilities and suitable candidates.

Due to the lack of harmonization of permitting and/or reporting requirements between regulatory agencies, the number of wildlife rehabilitators and the number of species and animals admitted for rehabilitation in the US on an annual basis are not known. Based on federal permit numbers, organizational memberships and a dated phone survey, there are several thousand wildlife rehabilitators [47,48,49]. Most wildlife rehabilitators are home-based or in a stand-alone facility and are self-supported; often, the access to veterinary services is limited, which can impact animal welfare. In a study to characterize wildlife rehabilitation across a seven-state area, one third of wildlife rehabilitators admitted fewer than 100 animals per year, and another third admitted fewer than 1000 animals per year [13]. Nevertheless, it is estimated that approximately 500,000 animals are admitted for rehabilitation annually in the US [1] (Ed Clark, Personal Communication, 2010). 

### 2.1. Federally Regulated Wildlife Rehabilitation

#### 2.1.1. Migratory Bird Rehabilitation

In the early 20th century, native bird populations were declining due to over-hunting, poaching and the millinery trade. In 1918, the Congress passed the Migratory Bird Treaty Act (MBTA), which officially made it a crime to “pursue, hunt, take, capture, kill,” or “sell” a migratory bird or any of its parts, including nests, eggs, and feathers [50]. The Bald and Golden Eagle Protection Act (BGEA) was enacted in 1940 to ban hunting and otherwise disturbing eagles [51]. In 1972, an amendment specifically added raptors to the MBTA. All these regulations serve to protect populations of birds.

The Code of Federal Regulations (CFR), Title 50 Fisheries and Wildlife, Chapter 1 United States Fish and Wildlife Service (USFWS), Department of the Interior defines and describes human activities related to migratory birds such as taxidermy, import/export, scientific collection, and rehabilitation. Covered birds include all wild birds native to North America, more than 1100 species [52]. Administered by the USFWS Migratory Bird Program, a permit category specifically to authorize migratory bird rehabilitation was created in 2003 [53]. Rehabilitation of migratory birds is defined as the practice of caring for sick, injured, or orphaned migratory birds with the goal of releasing them back to the wild [53].

CFR Title 50 Part 21.76 sets out migratory bird rehabilitation policies including age and experience of the applicant and enclosures based on criteria set out by the NWRA/IWRC *Minimum Standards for Rehabilitation* (3rd edition, 2000) [52,54]. If the applicant’s state requires a permit to possess migratory birds for rehabilitation purposes, the applicant must also comply with state requirements, which may be more restrictive. Applicants must also have an agreement with a licensed veterinarian to provide medical care for the birds.

A condition of a migratory bird rehabilitation permit is completion of an annual report [55]. This report requires information on species, numbers of birds, the nature of their injuries, and dispositions. It also requires reporting of potential criminal activity and optional reporting of confirmed infectious disease or toxicants. 

As to the scope of migratory bird rehabilitation, currently, the Migratory Bird Program has 1750 rehabilitation permits. While the total number of permits issued are tracked, the number of migratory birds, species admitted for rehabilitation and their outcomes are not tabulated except for eagles. The Raptor Center (TRC) looked at the 2011 annual reports for Region 3, which comprises eight states in the Midwest. Of the 1385 rehabilitation permits issued by USFWS that year, 254 permits were issued from Region 3. An analysis of 216 annual reports received showed that approximately 45,000 migratory birds were admitted for rehabilitation that year [48]. Another analysis of annual reports from seven northern states showed that approximately 25% of birds admitted were passerines, 10–15% were waterfowl species, and 10–15% were raptor species, with fewer numbers of doves, pigeons, and shorebirds [13]. Hanson reported that 12,863 passerines, 7015 pigeons and doves, 4081 waterfowl, and 3212 raptors were admitted to New York state rehabilitators between 2012 and 2014 [3].

#### 2.1.2. Sea Turtle Rehabilitation

There are seven extant sea turtle species, including green (*Chelonia mydas*), leatherback (*Dermochelys coriacea*), loggerhead (*Caretta caretta*), flatback (*Natator depressus*), hawksbill (*Eretmochelys imbricata*), olive ridley (*Lepidochelys olivacea*), and Kemp’s ridley turtles (*Lepidochelys kempii*). These species are globally threatened, having been negatively impacted by habitat loss, fishery interactions, vessel collision, pollution, disease, and unusual weather events [8,9,40,56,57,58,59,60,61,62] (Figure 1). Six of the seven sea turtle species are found within US waters, and all six are protected under the US Endangered Species Act of 1973 (ESA).

In the US, the USFWS and the National Oceanic and Atmospheric Administration’s National Marine Fisheries Service (NMFS, also known as NOAA Fisheries) share responsibility for sea turtle conservation [52,63,64,65]. As defined by NOAA, a “stranded” sea turtle is one that is found dead, or is alive but unable to behave normally due to an injury, illness, or other problem; and NOAA recognizes that stranding may occur on land or at sea [66]. Globally, thousands of sea turtles are found stranded each year, often due to anthropogenic causes, and many are already dead when found [67,68,69,70,71]. NOAA has legal authority over sea turtles at sea, while USFWS has legal authority over sea turtles on land. Therefore, USFWS legally oversees sea turtle rehabilitation, but NOAA is heavily involved operationally, as well. 

After sea turtles were ESA-listed, recovery plans were developed (e.g., NMFS and USFWS, 2019). The plans propose criteria by which sea turtle species are considered for delisting and describe activities that promote recovery. All the US sea turtle recovery plans recommend the existence of programs to manage stranded dead, ill, and injured sea turtles. The Sea Turtle Stranding and Salvage Network (STSSN) was formally established by NMFS in 1980 to document the strandings of sea turtles along the coastal US and the US Caribbean. NMFS employees participate directly in stranding response and transportation of stranded turtles in some situations and oversee mortality investigations. However, the number of NMFS employees who are assigned to this work is small relative to the number of strandings and associated resource needs. Thus, permitted private partners contribute most of the resources to respond to and document stranded sea turtles, determine causes of morbidity and mortality, and inform conservation management and recovery. While NMFS coordinates the networks, it is the participating local organizations that respond to stranded turtles, collect scientific data, transport sick and injured turtles to rehabilitation facilities, arrange veterinary care, and educate the public about sea turtle conservation. These facilities must follow the USFWS standards of care [72]. Annual reports regarding caseload and individual case data are required by USFWS, and in some circumstances, quarterly reports are required [65]. Sea turtles are also protected by the law in many individual states; thus, facilities must comply with both federal and state regulations.

As to the scope of sea turtle rehabilitation, as of March 2022, there were 58 facilities in 21 US states and territories permitted by USFWS to provide long-term veterinary care, rehabilitation, and release for sea turtles. Release rates vary among stranding events and regions, ranging from approximately 30 to 80% [67,68,69,70,71,73]. A recent survey indicated that sea turtle rehabilitation facilities had released approximately 11,000 turtles since their inception through 2016 [74]. 

#### 2.1.3. Marine Mammal Rehabilitation

In the US, marine mammals are federally protected under the Marine Mammal Protection Act (MMPA) and the ESA, and their captive care is directed under the AWA. For the purposes of this publication, marine mammals include species of the order Cetacea (whales, dolphins, and porpoises), Pinnipedia (seals, sea lions and walruses) and Sirenia (manatees and dugongs) and select species of the families Mustelidae (sea otters) and Ursidae (polar bears). The MMPA was enacted in 1972 and prohibits, with certain exceptions, the “take” of marine mammals in US waters and by US citizens on the high seas, and the importation of marine mammals and marine mammal products into the US. As amended in 1992, the MMPA established the Marine Mammal Health and Stranding Response Program (MMHSRP), commonly called the Marine Mammal Stranding Network. The MMPA makes it illegal for anyone except agents or authorized members of the response program to respond to marine mammals in need of care and stipulates that people stay at least 50 yards away from any marine mammal, whether ill or not. The ESA, enacted in 1973, provides a program for the conservation of threatened and endangered plants and animals and the habitats in which they are found. The law also prohibits any action that causes a “taking” of any listed species of endangered fish or wildlife. “Take” as defined under the ESA means “to harass, harm, pursue, hunt, shoot, wound, kill, trap, capture, or collect, or to attempt to engage in any such conduct”.

While the work of the MMHSRP is chiefly about the assessment, rescue and rehabilitation of stranded marine mammals, the effort advances scientific missions, allows for disease surveillance, provides for the early detection of such things as harmful algal blooms and serves as an incubator for advancing medical technologies, management, and conservation programs. Data amassed, and tissues cryopreserved, are the basis of research led by rehabilitation organizations and allied academic institutions. Furthermore, the care of patients that are part of healthy populations provides excellent models for the care of individual patients from populations of endangered species (such as the critically endangered vaquita and the Hawaiian monk seal). 

Management of sea otters and polar bears is overseen by the USFWS, and all other marine mammal species are overseen by NOAA Fisheries [75]. Certain species are afforded additional state or jurisdictional protections, such as select whale species and the Guadalupe fur seal through the California Endangered Species Act of 1970 [76], and the manatee, which is also protected by the Florida Manatee Sanctuary Act of 1978 [77]. (It is not the intent of this publication to cite every protection of every marine mammal beyond federal legislative protections.) In the US, NOAA’s MMHSRP provides the framework for a network of authorized marine mammal responder and rehabilitation entities including non-profit organizations, academic institutions, zoos and aquariums, municipalities, and individuals. The diversity among them is great in the response areas they cover, scope of activities, staffing, facilities and more, yet each is guided by the terms of an executed agreement with the agency, bringing a level of consistency to this work.

As to the scope of marine mammal rehabilitation, 2019 is the latest year for NOAA Fisheries’ national overview report. In 2019, there were 7719 confirmed marine mammal strandings in the US under NOAA Fisheries jurisdiction (i.e., marine mammals excluding polar bears and sea otters). These animals were responded to by more than 120 organizations, authorized and overseen by the same agency. This is slightly higher than the 13-year (2006–2018) average of 6365. Of these, 73% were pinnipeds (comprised mostly of California sea lions, harbor seals, northern elephant seals, gray seals, and harp seals, in descending order), 23% were small cetaceans (comprised mostly of common bottlenose dolphins, harbor porpoises, short-beaked common dolphins, Atlantic white-sided dolphins, and long-beaked common dolphins, in descending order), and 4% were large whales (comprised mostly of gray whales, humpback whales, minke whales, bowhead whales, and fin whales) [78].

### 2.2. State Regulated Wildlife Rehabilitation

#### Terrestrial Wildlife Rehabilitation

Regulations on dealings with terrestrial animals, including wildlife rehabilitation, are determined by the states and can vary considerably. In some states, wildlife rehabilitation is not permitted on any animal, and in others, it may be limited to only native species or “non-nuisance” species. When wildlife rehabilitation of native terrestrial animals is allowed, establishments such as state departments of fish and game, natural resources, animal health and/or agriculture oversee these activities and often have permit requirements. Accessing a state’s website is the best way to determine its requirements. Requirements may include submitting an application, paying a fee, passing a test, and/or inspection by a wildlife official such as a conservation officer. In some cases, a specialized certification may be required for some species, such as large carnivores and cervids.

If regulation does not specify if and/or when invasive or non-native species may be rehabilitated, it is often left to the individual centers or rehabilitators to decide whether they will accept particular species for care, euthanasia or not at all. The NWRA/IWRC Wildlife Rehabilitator’s Code of Ethics, the facility’s mission, and/or the rehabilitator’s value system may determine policies regarding invasive and non-native species [79,80]. 

The rehabilitation of rabies vector species (RVS) may also be influenced by the area’s department of health. Rabies vector species vary by region based on the prominent viral strain in the area; the most prevalent strains in the US are the raccoon, skunk, bat, and fox strains. When a RVS species is allowed to be rehabilitated, focused training and facility inspection are often required to protect the wildlife rehabilitators working with these animals. For wildlife rehabilitators working with RVS, the Centers for Disease Control and Prevention (CDC) recommends rabies pre-exposure prophylaxis (PrEP) [81]. The emergence of new infectious diseases and zoonoses such as SARS-CoV-2 and Highly Pathogenic Avian Influenza may also alter which species can be rehabilitated and where.

Venomous and poisonous herptile species are not commonly presented for care, but several species exist in North America. Thus, a rehabilitator working with reptiles and amphibians must be able to identify a potentially venomous or poisonous animal and possess the skill and personal protective equipment to safely handle and treat these animals or have a resource for someone who can. The NWRA retains a small list of wildlife rehabilitators who possess permits to work with these animals. 

While protection of human health and animal agriculture are important considerations in wildlife rehabilitation, these policies can create an issue with how best to handle species that cannot be admitted for rehabilitation. Often these animals end up in untrained hands with poor outcomes for the animals and the public.

As to the scope of terrestrial wildlife rehabilitation, based on federal permit numbers, organizational memberships and a dated phone survey, there are several thousand wildlife rehabilitators in the US [47,48,49], and many of them work with terrestrial animals. Nearly half of all animals admitted for rehabilitation are considered terrestrial mammals (including bats), reptiles, and amphibians. Mammals, and particularly lagomorphs and rodents, comprise the largest percentage of terrestrial animals, although the total percent and the different species totals vary slightly by region. A recent three-year retrospective analysis of nearly 20,000 wildlife patients from three large centers in Canada found that of all admits, 40.7% were mammals, 1.2% were reptiles and 0.05% were amphibians [4]. Similarly, a three-year survey in the state of New York found that of the 58,185 wild animals admitted for rehabilitation, 43.7% were mammals, 4.2% were reptiles, and 0.1% were amphibians [3]. 

## 3. Standards of Care in Wildlife Rehabilitation

In wildlife rehabilitation, there are two major categories of standards of care: policy requirements promulgated by regulatory agencies and best practices promulgated by professional groups. The permitting federal agencies and most of the permitting state agencies have minimum requirements and/or policies for rehabilitation that must be adhered to as a condition of the permit issuance (see above). The rigor of these requirements or policies varies widely and can be obtained from the appropriate regulatory authority. States may be more restrictive, but not less restrictive than federal requirements and/or policies. 

Federal and most state regulatory agencies require rehabilitators to have a “relationship” with a licensed veterinarian (veterinarian of record (VOR)); again, the rigor of this relationship varies widely. Licensed veterinarians are not required to obtain a federal or state rehabilitation permit to euthanize wildlife or temporarily possess and stabilize wildlife. However, within 24–48 h of stabilization, the veterinarian should transfer the patient to a permitted rehabilitator. Other federal and state acts must be adhered to during the process of wildlife rehabilitation regarding the handling and use of controlled substances and the prescribing and use of anesthetics, antimicrobials, vaccines, hormones and other substances [82]. This is important to consider in wild species that may be hunted for food such as birds, rabbits, deer, bears, etc. [83].

Not all wildlife admitted for rehabilitation require veterinary intervention. For example, healthy, displaced juveniles generally do not require veterinary care. In a study to characterize wildlife rehabilitation across seven northern states, the majority of rehabilitators said that a veterinarian saw 25% or less of the animals admitted; only 15% said that all animals admitted were seen by a veterinarian [13]. The majority of rehabilitators felt that the admitted animals did not require a veterinarian, but 50% also said that a veterinarian was not accessible or available, or that expense was a barrier. Ideally, all sick or injured animals, or specialized taxa, such as sea turtles and marine mammals, should have veterinary support. Each state legally defines the practice of veterinary medicine within its borders. The American Veterinary Medical Association (AVMA) defines the practice of veterinary medicine as “to diagnose, prognose, treat, correct, change, alleviate, or prevent animal disease, illness, pain, deformity, defect, injury, or other physical, dental, or mental conditions by any method or mode,” which would encompass a great deal of practices within wildlife rehabilitation [16]. The AVMA Model exempts “any person who lawfully provides care and rehabilitation of wildlife species under the supervision of a licensed veterinarian.” However, the AVMA definition of supervision is quite precise and presumes the veterinarian “has assumed responsibility for the veterinary care given to the patient”.

As for best practices, the IWRC and NWRA jointly established *Standards for Wildlife Rehabilitation*. Currently in its 5th edition, this publication sets standards of care for ill, injured, and displaced wildlife [44]. While professional standards of care are not considered legally binding, some regulatory authorities may use them as such. 

Wildlife rehabilitation continues to evolve into two distinct phases, the veterinary phase and the reconditioning and release phase. The veterinary phase, or clinical wildlife medicine, is a developing specialty within the veterinary field for the diagnosis and treatment of individual, free-living wildlife [32,36,62,84]. Rehabilitation, or the reconditioning and release phase, is the pre-release physical and psychological preparation of animals, optimizing their ability to survive in the wild [85]. Both veterinarians and rehabilitators continue to define and develop their respective evidence-based science which informs best practices.

The primary goal of rehabilitation is to release a healthy animal back to the wild that can survive and thrive in its natural habitat at the same level as its conspecifics. The prognosis for the animal to be restored to this level of health should be the major consideration in the decision to treat to release. When there is no realistic expectation for this outcome, a secondary option, euthanasia or captive placement, is considered.

Some wildlife rehabilitation facilities have a policy to only “treat to release”; such facilities do not “treat to place.” This policy may be based on ethical grounds or on a lack of mechanisms to place animals [86]. It should also be noted that regulatory authorities may place further restrictions on specific outcomes in general or by species. For example, some agencies do not allow any captive placement and some agencies may require euthanasia of certain clinical conditions, invasive species, RVS or other public health or animal agriculture threats, or individuals such as male cervids (deer species) or venomous herptiles that present a danger to humans [45]. 

### 3.1. Standards of Care in Migratory Bird Rehabilitation

Beginning in 1989, CFR Title 50 Part 13.41 set forth that “any live wildlife possessed under a (USFWS) permit must be maintained under humane and healthful conditions” [87]. CFR Title 50 Part 21.76 sets the standards for migratory bird rehabilitation which include permit requirements and general permit provisions, permit application requirements and criteria, permit standard conditions, application to oil and hazardous waste spills, additional state requirements, and length of permit term [88]. A permit authorizes a rehabilitator to take from the wild or receive from another person sick, injured, or orphaned migratory birds and to possess them and provide rehabilitative care for them for up to 180 days. The permittee is also authorized to transfer, release, or euthanize such birds. Additionally, the permittee can transport birds as needed for release to another permitted rehabilitator’s facilities or to a veterinarian.

The majority of the standards of care fall under Standard Conditions (e) and include facilities, dietary requirements, disposition of birds, and reporting and record-keeping requirements. Reporting requirements include notification of the USFWS if the rehabilitator has reason to believe that a bird has been poisoned, electrocuted, shot, or otherwise subjected to criminal activity. Notification of state or local authority is required if sickness, injury, or death of any bird occurs due to an avian virus or other contagious diseases or public health hazards if an agency is currently collecting such information. Unfortunately, most wildlife rehabilitators lack the resources for laboratory testing for toxicities or disease or radiology for projectiles, resulting in significant underreporting of these conditions [89].

Federal regulators have additional policies regarding imprinting or habituating birds, federally threatened and endangered migratory bird species, and specific requirements for the disposition of dead birds, their parts, and feathers. There are also additional policies regarding outcomes, including temporary captive placements for the purpose of fostering juveniles or with permitted falconers for the purpose of reconditioning.

Outside of regulatory policies, there are a variety of sources for avian best practices. Many professional organizations have guidelines, standards of care, and/or publish evidence-based science. In addition to *Standards for Wildlife Rehabilitation*, the NWRA publishes *Principles of Wildlife Rehabilitation* [90]. Both the NWRA and IWRC and other rehabilitation organizations offer in-person and online training related to avian rehabilitation as well as publications. The AZA has Animal Care Manuals for a variety of avian species or taxonomic groups, as does the Global Federation of Animal Sanctuaries [42,91]. The American Association of Zoo Veterinarians (AAZV) and the Association of Avian Veterinarians (AAV) present and publish materials applicable to medical conditions seen in birds undergoing wildlife rehabilitation, as well as how to assess avian welfare.

### 3.2. Standards of Care in Sea Turtle Rehabilitation 

Veterinary management and rehabilitation of sea turtles has become quite sophisticated, and details of medical and surgical management of various conditions have been described [92]. Recommendations of expert technical panels and models of prognostication and mortality prediction have been created [93,94]. Indeed, in the US, sea turtle rehabilitation facilities must have access to a veterinarian who has experience with sea turtles, or, minimally, the attending veterinarian must have a written consulting agreement with an experienced veterinarian. The attending veterinarian must also be available at all times, such that each patient can be examined within 24 h of admission [65]. 

In the rehabilitation facility, sea turtle wellbeing is maintained by providing adequate space, appropriate temperature, consistent light cycles, excellent water quality, excellent nutrition and dietary variety, excellent medical care, and environmental enrichment. Clearly, one cannot reproduce the freedom and habitat complexity of the oceanic environment for such highly migratory and athletic species [72]. However, minimum standards for enclosure size, temperature, water quality, and diet have been published [65]. For example, in the US, among other requirements, sea turtles must be maintained in water with salinity between 20 and 35 ppt unless otherwise directed by a veterinarian, and water total coliform concentrations must be kept below 1000 MPN per 100 mL [65]. Rehabilitation pools for sea turtles are best designed with advanced life support systems, including filtration and disinfection, that reduce the need for “dump-and-fill” management, which increases animal handling and presumably stress. Stocking density should be reduced as much as possible to reduce social stress and traumatic injury (e.g., conspecific biting is common). Enrichment should be provided in the form of a varied diet, diet presentation, “cleaning stations” (where turtles can achieve physical contact and light abrasion that they seem to seek out), and non-ingestible toys. Welfare assessments may include indicators such as appetite, use of enclosure space, diving capability, time spent in appropriate and inappropriate resting postures, body condition, medical status, and conspecific interactions. Permission is required to pursue research involving rehabilitating sea turtles (e.g., pharmacokinetic studies), and wellbeing must be ensured. Research study designs and methods must be approved by legal authorities and institutional animal care and use committees [65]. 

Sea turtle rehabilitation programs have the same disposition options as those relevant to other taxa. In almost all scenarios, sea turtles are released to the wild, which offers them a chance to reproduce and contribute to the population of their species. Euthanasia and permanent captivity remove the turtles from the reproductive population; thus, these options are only elected in extreme circumstances as discussed in Section 4.2 below. The USFWS must approve euthanasia and permanent captivity for sea turtles.

### 3.3. Standards of Care in Marine Mammal Rehabilitation

Only entities that hold a letter of authorization from NOAA Fisheries may rehabilitate marine mammals that fall under the auspices of the MMPA. The care of these species is guided by the 2022 NMFS Standards for Rehabilitation Facilities [75]. The captive care of marine mammals is directed under the AWA, and while only a fraction of marine mammal rehabilitation facilities conduct research, for these organizations, the AWA also applies. 

A marine mammal “strands” when it is out of its normal habitat. Whales, dolphins, and porpoises (cetaceans) are considered stranded when they are found dead, either on the beach or floating in the water, or alive on the beach and unable to return to the water. Seals and sea lions (pinnipeds) are considered stranded when they are found dead on land or in the water, or are in need of medical attention. Because healthy pinnipeds come on land to rest, expert assessment may be needed to determine whether they need help. Live-stranded animals usually need medical attention or professional assistance to return to their natural habitat [95]. 

Marine mammal rehabilitation has the same dispositions as non-marine mammal wildlife rehabilitation. While rehabilitation time varies, a patient may not be kept in rehabilitative care for more than 6 months without specific authorization. NOAA Fisheries and a veterinarian must approve a release to the wild or a captive placement. Euthanasia must be guided by AVMA guidelines.

There exists a robust body of literature around rehabilitative care of marine mammals, namely pinnipeds, small cetaceans, and otters. Rehabilitators are referred to 2022 NMFS Standards for Rehabilitation Facilities [95,96,97]. In addition, the AAZV (https://www.aazv.org/, accessed on 5 April 2023), the International Association of Aquatic Animal Medicine (IAAM) (https://www.iaaam.org/, accessed on 5 April 2023) and the AZA (https://www.aza.org/, accessed on 5 April 2023) are offered as sources of information.

Guidelines are provided and legal and acceptable measures are allowed for the safe “hazing” of healthy animals from areas where they pose a threat to themselves, people, other animals, or property. Relocating marine mammals, such as those that may be hauled out on docks in busy marinas, may well avert harm or injury, and prevent the need for active rehabilitation. This is a complicated reality for rehabilitators who are called on to care for ill or injured marine mammals and who also may be sought when marine mammals are seen as nuisances. The best response in such instances is for the responder to refer the inquiring party to NOAA’s website for deterring nuisance pinnipeds (https://www.fisheries.noaa.gov/west-coast/marine-mammal-protection/deterring-nuisance-pinnipeds (accessed on 15 April 2023)).

### 3.4. Standards of Care in Terrestrial Wildlife Rehabilitation

*The Principles of Wildlife Rehabilitation* and *Standards for Wildlife Rehabilitation* established by NWRA and IWRC as well as *Medical Management of Wildlife Species: A Guide for Practitioners* are good resources for rehabilitating wildlife and help to establish best practices in the sector [36,90]. Both the NWRA and IWRC and other rehabilitation organizations offer in-person and online training as well as species or taxon-specific publications. Besides IWRC and NWRA, there are approximately 26 additional state wildlife rehabilitation organizations throughout the US. These organizations are dedicated to the training and support of wildlife rehabilitators in their state.

Terrestrial mammals have the same dispositions as other species in need of rehabilitation. For wild animals that do require care from a rehabilitator, the process of rehabilitation varies widely. Most terrestrial wildlife species admitted for care, such as cottontail rabbits, Virginia opossums, various species of squirrels, and reptiles including painted and box turtles, are commonly seen in urban and suburban areas. Their coexistence with humans in urban areas makes them highly susceptible to some of the most common anthropogenic reasons for admission of wildlife, including infectious diseases from backyard feeders, vehicle collisions, window strikes and cat attacks, all of which are familiar to many rehabilitators [6,7]. 

Some less commonly encountered species require particular training with specialized enclosures and medical care. For example, large carnivorous animals such as ursids and felids require ample space and should be managed as “hands-off” as possible to reduce the risk of imprinting and habituation, which can pose a severe risk to the animal and humans after release. Additionally, orphaned carnivores must demonstrate the ability to recognize, capture and kill appropriate prey before release [44]. An extensive review of the unique needs of various species in wildlife rehabilitation settings is beyond the scope of this paper but remains a critical component of providing appropriate, effective, and efficient care to each animal admitted for rehabilitation.

Regardless, due to the variety of species and levels of rehabilitation, there is a need to develop detailed standard operating procedures for common and uncommon species and conditions based on evidence-based science and post-release monitoring. 

## 4. Welfare Considerations during Wildlife Rehabilitation

It is likely that the humans who have cared for injured and orphaned wildlife over the decades have also cared about the “welfare” of these animals even before the concept of animal welfare was defined. The construct for animal welfare began in the United Kingdom in 1965 with the Five Freedoms [98] and has since developed into the Five Domains [99,100] and the Three Spheres [101]. Additionally, some AZA facilities have adopted the Five Opportunities to Thrive [102]. The definition of animal welfare varies among organizations and professional societies. To name but a few, the World Organisation for Animal Health (WOAH, previously the OIE), the American College of Animal Welfare (ACAW), the AVMA, and AZA all have similar but targeted definitions [16,17,103,104]. Recently, the AZA made a distinction between animal “welfare” and “wellbeing,” with animal welfare defined as the scientific study of the welfare of animals, and wellbeing defined as “the state of being comfortable, healthy, or happy” [103]. Regardless of the definition or term, it is important to recognize that welfare resides within the individual animal; it is not given to them. It is a measure of how the animal is coping with the conditions in which it lives—its feelings, behavior, and health [17,105]. The animal’s welfare may be negative, neutral, or positive, and it varies from moment to moment [106]. The aggregate balance of positive and negative states implies a quality of life and connotes a life worth living [107]. 

Juvenile or adult, predator or prey, healthy or unhealthy, the process of rehabilitation has a significant impact on any wild animal admitted for rehabilitation, a life interrupted. Societal benefits aside, the only benefit to the animal is to relieve its suffering and, if possible, restore the path of its natural life. Deciding if and how to do so requires a continuing series of deliberate decisions, using a complex amalgam of objective and subjective inputs, to optimize the animal’s welfare outcome (Figure 2).

Regulatory policies impact initial decisions made by wildlife rehabilitators and veterinarians. While these federal, state, and occasionally local policies are often the most objective inputs a rehabilitator or veterinarian has, they are sometimes poorly defined and cannot address every situation. Next, decisions are informed and guided by any available standards of care and professional best practices. Unfortunately, there is often limited evidence-based science in wildlife rehabilitation and wildlife medicine. Finally, decisions are influenced, consciously or unconsciously, by ethics [108]. While there are codes of professional conduct for both wildlife rehabilitators and veterinarians, often, decisions are made based on a core sense of ethics and personal values [16,79].

There have been many attempts to apply foundational ethical theories, such as utilitarianism or deontology, to animals, or to develop uniquely animal-centric theories such as that posed in A Practical Ethics for Animals by Fraser [109]. In this article, Fraser outlines the ways in which humans affect animals: keeping of animals, intentional harm, direct but unintended harm, and indirect harm by disturbing life-sustaining processes and balances of nature. Fraser proposes four mid-level principles to address the main ethical concerns: provide good lives for the animals in our care, treat suffering with compassion, be mindful of unseen harm, and protect the life-sustaining processes and balances of nature. 

Subsequently, Fawcett applied Fraser’s work to a variety of veterinary practices, including wildlife rehabilitation [110]. Fawcett further developed and determined the compatibility of A Practical Ethics for Animals with a One Welfare framework. Health professionals are becoming increasingly familiar with the concept of One Health—the interconnectedness of the health of people, animals, and the environment—and the necessity for an interdisciplinary approach in these fields [111,112,113]. This more recent concept, One Welfare, recognizes the many interconnections between human wellbeing, animal welfare, and the integrity of the environment [114,115].

Human activities are having an ever-greater influence on wildlife. This impact is often cited as the reason for wildlife rehabilitation, namely that it is our moral responsibility. Wildlife rehabilitation is reactive. A proactive approach, working “upstream” to prevent wild animals from needing rehabilitation, would be the best way to improve their welfare, and ours [2,3,116].

### 4.1. Migratory Bird Welfare

Federal policy has limited welfare standards for migratory birds. Wildlife must be maintained under humane and healthful conditions [87]. Additional conditions are set for physical facilities and dietary requirements, as well as the different outcomes in migratory bird rehabilitation [88]. 

Wildlife rehabilitators are the first line of defense in preventing juvenile birds from being “chick napped” and brought to rehabilitation facilities unnecessarily, as well as mitigating the human–wildlife conflict and educating the public regarding policies involving migratory birds [50]. Every effort should be made to reunite and/or renest healthy or successfully treated juvenile birds with natal or conspecific foster parents (Figure 3). Facilities should maintain a database of area nest locations by species for this purpose. Post-release monitoring should occur whenever possible, along with appropriate documentation [44]. 

Release, euthanasia, and captive placement rates vary widely between avian rehabilitators, and comparisons should be avoided. A facility that admits large numbers of displaced juvenile waterfowl should release a large percentage of them. A facility that admits large numbers of raptors due to trauma should euthanize a large percentage of them for humane reasons [2,117,118]. Avian rehabilitators, along with their VOR, should design facility-specific policies for euthanasia, release to the wild, and captive placement for the species or taxonomic groups admitted (Table 1).

#### 4.1.1. Euthanasia of Migratory Birds

If the best course of action is humane euthanasia, an avian rehabilitator must obtain permission from the USFWS before euthanizing a threatened or endangered bird species, unless prompt euthanasia is warranted due to humane consideration for the welfare of the bird [88]. This holds true for veterinarians as well; they are permitted a wide latitude to euthanize animals for humane reasons without prior permission. Euthanasia should still be reported afterward. Rehabilitators must euthanize any bird that cannot feed itself, perch upright, or ambulate without inflicting additional injuries to itself, where medical and/or rehabilitative care will not reverse such conditions. Permittees must, with few exceptions, also euthanize any bird that is completely blind and any bird that has sustained injuries that would require the amputation of a leg, foot, or wing at the elbow or above (humeroulnar joint) rather than performing such surgery [88]. 

These are minimal criteria for euthanasia based primarily on physical or physiological welfare states, with less regard for affective or behavioral welfare states [101]. The criteria are also trauma-centric. While trauma is a common reason for birds to be presented for rehabilitation, presentations increasingly include disease, toxicity, and failure to thrive. Additionally, the criteria are applied to all species without considering individual species or taxonomic groups. The amputation of a wing below the elbow in a long-lived, heavily bodied species that perches upright may have a different impact than in a lightweight, horizontally aligned species that spends the majority of its time on water. Ideally, these migratory bird euthanasia criteria should be periodically reviewed, updated, and strengthened. As the literature in avian welfare, avian medicine, and wildlife medicine advances, additional criteria should be considered for inclusion in triage euthanasia. For example, luxations, immobile joints, and intra-articular fractures commonly lead to degenerative joint disease, a painful condition that should preclude release to the wild or captive placement [119,120,121]. A comparison of rehabilitation rates of birds of prey from a raptor rehabilitation center 10 years apart showed that over time, higher numbers of birds were euthanized, and fewer birds were released [122]. Anecdotal evidence suggests that a legislative push to decrease the duration of stay on welfare grounds may have led to a quicker decision-making process, or a lower threshold criterion for euthanasia.

Finally, migratory bird policy has no guidelines regarding how birds should be euthanized. Ideally, all animals should be euthanized based on the most current recommendations of the AVMA [123]. A survey was distributed to wildlife rehabilitators [124]. Ahlmann-Garcia (2021 unpublished) found that rehabilitators, not veterinarians, euthanized the majority of patients at facilities. Dysthanasia, a euthanasia attempt with an undesirable outcome, may commonly occur in non-domestic species [125]. While most rehabilitators in this survey met AVMA standards for euthanasia, veterinarians have more resources available for humane euthanasia and methods of confirming death. Unfortunately, not all rehabilitators have access or timely access to veterinarians. Avian rehabilitators should work with their veterinarian to design a facility-specific euthanasia policy for species that are admitted.

#### 4.1.2. Release of Migratory Birds to the Wild

The broad diversity of avian species admitted for rehabilitation precludes specific details of care, and appropriate avian, veterinary and wildlife rehabilitation best practices should be consulted (see Section 3.1). Considerations include nutrition, environment, veterinary care, ability to express species-specific behaviors and stress mitigation. Another consideration is time. Rehabilitation is not without its consequences. Birds see humans as predators, which makes being cared for an innately stressful experience. The safety and efficacy of many drugs, including analgesics, is limited or unknown for most avian species. Rehabilitators and veterinarians need to constantly re-evaluate each bird’s physical, emotional, and behavioral prognosis. Is the bird still releasable, and what are the resources required? (Figure 4).

Avian rehabilitators must take every precaution to avoid imprinting or habituating birds. If a bird does become imprinted while under care, the rehabilitator is required to transfer the bird to another facility [88]. This approach encourages the best chances of successful release and discourages deliberate malimprinting [126] for the purposes of obtaining an educational or an ambassador animal.

Environmental and behavioral enrichment is an important component of stress reduction. There is little literature on environmental enrichment for wild birds undergoing rehabilitation, and these settings are much more challenging than for birds in permanent captivity. Just as important as providing enrichment is critically evaluating the result of the enrichment to ensure that it is having a positive impact [127,128].

Once a bird has attained a condition suitable for release, the rehabilitator must release it to suitable habitat as soon as seasonal conditions allow. Migratory birds cannot be held longer than 180 days without additional USFWS authorization [88]. This authorization is often sought by northern avian rehabilitators as releasable birds may need to be overwintered before weather conditions are favorable for release. Finally, before releasing a threatened or endangered migratory bird, the rehabilitator must comply with USFWS requirements [88]. As noted above, states may have additional avian rehabilitation requirements regarding the release of birds, as well as their own lists of threatened or endangered species.

The hazards of release are generally underestimated. Avian pre-release protocols should include general and species-specific criteria, as well as considerations based on the bird’s individual history. Anatomical, physiological, biological, behavioral, and environmental criteria and standards should be defined [44,84]. A proximate pre-release health and behavior evaluation should be performed. A thorough eye exam should always be performed. If the bird’s history includes any joint or bony trauma, final radiographs should be performed to ensure proper healing and alignment and absence of degenerative joint disease.

##### Post-Release Monitoring of Migratory Birds

Post-release monitoring of rehabilitated birds can indicate whether the treatment and reconditioning phases of rehabilitation were successful, and whether rehabilitation has any long-term impact on the individual, their populations, or the environment. There are three main strategies for post-release monitoring: visual observation, banding, and telemetry. Visual observation has limited practicality in aerial species and is most effective when used in conjunction with other strategies. Banding is regulated by the US Geological Survey (USGS). On average, each year, 1.2 million birds are banded, and 87,000 encounters are documented. An “encounter” is any observation of a previously banded bird, living or dead. Banding is inexpensive and relatively passive. Unfortunately, the encounter rate of rehabilitated birds is low (5–8%), requiring a large number of birds to obtain statistically significant results. The third strategy for post-release monitoring is radio telemetry that uses Very High Frequency (VHF) transmitters or the Global Positioning System (GPS), cellular or satellite tracking. Telemetry allows for more detailed monitoring, but has a variety of limitations including cost, loss of the transmitter, and technological malfunctions. The transmitter must not impair any of the bird’s activities. Unfortunately, there is limited data from post-release monitoring of rehabilitated birds. The majority of articles come from oiled bird studies because some of the monetary damage assessments to the responsible party can be used for post-release monitoring [129,130,131]. Other studies have looked at post-release survival for a variety of injuries or the success of hand-reared or captive raised fledglings or adults [132,133,134,135]. Overall, post-release studies are limited in number and depth, with various definitions of success, or no definition at all [136]. 

#### 4.1.3. Captive Placement of Migratory Birds

Avian rehabilitators may place non-releasable live birds that are suitable for use in educational programs, foster parenting, research projects, or other permitted activities with persons permitted or otherwise authorized to possess such birds with prior USFWS approval [88]. Whenever migratory birds are being placed into permanent human care, the bird must be transferred from the facility with the rehabilitation permit to the new facility with the appropriate permit [137]. For use in educational programming, such as for exhibition or as an ambassador animal, a Migratory Bird Special Purpose Possession—Education (Live) permit is required. Prior to authorizing the transfer, the USFWS requires a veterinarian’s statement and enclosure and experience information. There is a fillable template for the veterinarian’s statement (https://www.fws.gov/migratorybirds/pdf/policies-and-regulations/templatevetletter.pdf, accessed on 28 March 2023). The bird must be examined by the veterinarian within 30 days of the proposed transfer. The veterinarian’s statement must fully describe the medical or behavioral condition of the bird, why it renders the bird non-releasable to the wild, and why the bird is suitable for life in captivity. If the bird is being transferred, the veterinarian must provide the name of the veterinarian or animal care professional whom they consulted with at the receiving facility.

If the bird’s conditions meet the euthanasia criteria above (blind or has amputations), but the bird is not euthanized, there are additional requirements. First, the veterinarian must explain why the bird is not expected to experience long-term complications in captivity associated with the above-described injuries and/or ailments. Second, the veterinarian must express a commitment to provide medical care for the bird for the duration of its life, including complete examinations at least once a year, or, if the bird is being transferred, the name of the veterinarian who will provide such care. Third, an appropriate placement must be available. Lastly, the USFWS must specifically authorize the continued possession, medical treatment, and rehabilitative care of the bird [88]. 

It is unclear from 50 CFR 21.76 how this process should proceed in acute medical situations. Often, a decision is made by a rehabilitator and/or a veterinarian to proceed with medical or surgical treatment that will render the bird non-releasable prior to determining the bird’s emotional suitability for captivity, placement availability, or authorization from the USFWS.

Captive placement has the most regulatory requirements, as it should, because it has a significant impact on animal welfare. The natural life span of many avian species runs into decades or longer in captivity. Captive placement of an individual with physical or behavioral issues can lead to a poor quality of life for a lifetime. Many non-releasable birds are placed in zoos or nature centers, either as exhibit birds or for use as ambassador animals. Live ambassador animals are thought to engender public empathy, and by doing so, help promote ways in which people can act in relation to wildlife. It is likely that more birds are captively placed than any other rehabilitated taxa. The Raptor Center evaluated the placement of native birds via the 2017 Education (Live) permits. Nationally, there were 701 Live permits, with 162 in Region 3 (Upper Midwest). The Region 3 permits contained 1250 birds of which the vast majority were raptors. These birds came almost exclusively (89%) from dedicated wildlife rehabilitation organizations. It should be noted, however, that there are hundreds of facilities (such as AZA accredited zoos, academic institutions, municipal facilities, etc.) that are exempt from these permits, except in the case of eagles, so the actual number of native bird placements on an annual basis is much higher [89].

Beginning in February 2024, the AWA will expand to include some avian species. Wildlife rehabilitators who also maintain birds under the USFWS Migratory Bird Special Purpose Possession—Education (Live) and who have more than four raptors for exhibition will be required to apply to USDA APHIS AC for an exhibitor’s license [138]. 

Not all regulators allow for captive placement and not all facilities engage in captive placement, often based on ethical grounds. Disabilities that lead to a non-releasable status can also preclude a high quality of life in human care. Increasingly, avian facilities are only accepting those birds with the physical, psychological, and behavioral capacity to have a life-long high-quality welfare [119,120]. 

The International Association of Avian Trainers and Educators (IAATE) recently published a position statement titled Selection Considerations for Non-releasable Birds that states that “at a minimum, non-releasable birds used in education should be able to eat on their own, maintain good feather condition with regular preening, bathing, and the absence of fractious behavior, exhibit species-appropriate behaviors, have full mobility, and willingly engage with trainers” [121]. The position statement includes several pages of acceptable and unacceptable disabilities along with the reasons behind the designations. 

The decision to place a bird should be made after rehabilitation for release to the wild has failed; except for very rare circumstances, birds should not undergo rehabilitation for the purpose of placement. All medical issues should be resolved, and a current physical exam performed to ensure that the bird’s physical condition is stable and will not result in any pain or discomfort. Next, the bird should be evaluated by an experienced avian behaviorist to determine emotional and behavioral adaptability to captivity. Finally, a suitable facility with all the resources required to support this individual bird must be available [85,119]. Birds should not be “warehoused” while waiting for placement. If all these criteria cannot be met, the bird should be humanely euthanized.

There are best practice resources for non-releasable birds placed into educational programs. The NWRA published a book titled *Wildlife in Education: A Guide for the Care and Use of Program Animals* [139]. In addition, the NWRA has a Wildlife Educator’s Code of Ethics [80]. The AZA has a variety of resources including an Ambassador Animal Evaluation Tool (https://assets.speakcdn.com/assets/2332/ambassador_animal_evaluation_tool.pdf, accessed on 28 March 2023) and the Ambassador Animal Resource and Information Center (https://ambassadoranimalsag.wordpress.com, accessed on 28 March 2023). IAATE also has resources (https://iaate.org, accessed on 7 April 2023).

### 4.2. Sea Turtle Welfare

An excellent review of welfare considerations for sea turtles in human care was recently published [72], and a number of studies have evaluated enrichment methods for sea turtles (e.g., Escobedo-Bonilla, et al., 2022) [140]. The relatively extensive laws and regulations for sea turtle rehabilitation generally result in a high level of wellbeing for sea turtles under human care. It is rare, for example, for an unauthorized good Samaritan to attempt to care for an injured or ill sea turtle in a home setting. Nonetheless, several welfare considerations exist during initial recovery of a stranded turtle, proceeding through medical care, rehabilitation, and eventual release. 

The welfare of sea turtles in general will be improved by ongoing efforts to reduce anthropogenic injury and illness. If interactions by boat strikes and fisheries could be reduced, many fewer sea turtles would require hospitalization. Technological methods such as turtle excluder devices and ropeless fishing technology offer hope; public outreach methods in certain regions at certain times (e.g., during nesting season) could reduce boat-strike injuries [8,9]. 

During initial recovery of sea turtles from a stranding location, welfare considerations include prevention of further injury and pain, provision of proper environmental temperature, limitation of physiologic stress, and efficient transfer to an authorized rehabilitation center. In general, only personnel from permitted agencies should physically interact with a stranded sea turtle, although good Samaritan interventions may occur. Initial efforts may include moving a turtle away from drowning risk and away from temperature extremes and intense sunlight. Responders must be cautious to support injured appendages and shell fractures to prevent further injury and pain. Planks of wood, blankets, etc., may be used cautiously to support the turtle from below to relocate it. In some cases of “debilitated turtle syndrome,” turtle bodies may become so fragile that they are at risk for catastrophic injury during handling (e.g., carapace fracture causing visceral laceration) and extreme care is needed [72,94]. Temperature management is important for these ectothermic species, and in general, sea turtles should be held between 21 and 27 °C (70–80 °F) [65]. However, turtles that have stranded due to low environmental temperatures (cold-stunning) should not be warmed quickly, as it may lead to further physiologic compromise [67,72,92]. Finally, turtles should be transported efficiently, out of water, and with minimal stimulus [72]. It is clear that stranding and transport are physiologically stressful for sea turtles [61,141], and all efforts to eliminate transport duration, vibration, noise, and visual stimuli are warranted. 

#### 4.2.1. Euthanasia of Sea Turtles

Several conditions may lead to a decision to euthanize a sea turtle or to retain it under permanent human care, including spinal injuries, total blindness, multiple limb amputations, chronic buoyancy disorders, and severe fibropapillomatosis. Consent for euthanasia or permanent human care must be acquired from official regulatory agencies (such as USFWS). When euthanasia is elected, multimodal methods are recommended, generally including general anesthesia, followed by intravenous pentobarbital, and one or more adjunctive methods such as intrathecal or intracranial lidocaine, and intravenous or intracardiac potassium chloride (to end cardiac function that can otherwise often persist for hours) [92,142,143]. In certain scenarios (e.g., at sea), and less desirably, methods of exsanguination, decapitation, or other neuro-destructive methods may be required (e.g., captive bolt) [92]. Unless fresh tissues are needed for post-mortem investigations, a final step of freezing, after completion of other methods, ensures cessation of vital function. 

#### 4.2.2. Release of Sea Turtles to the Wild

Upon completion of rehabilitation, the wellbeing of sea turtle patients is enhanced by efficient release to the wild. USFWS requires sea turtles be examined by a veterinarian to ascertain good health prior to release, and then, unless other circumstances prevent it, turtles must be released within two weeks of medical clearance [65]. Delayed release is common, however, due to seasonal environmental conditions (e.g., water temperature being too cold in winter in certain regions). During such delays, which may often last up to 6 months, it is imperative to use the methods described above to maintain wellbeing. Depending on location, time of year, and caseload, turtles may need to be transported over long distances to be released at appropriate sites. The physiologic effects of long-distance vehicle transport have been studied for rehabilitated Kemp’s ridley and loggerhead turtles [144,145]. Physiologic stress was demonstrated, including elevation of plasma corticosterone, blood glucose, and heterophil to lymphocyte ratio, while other general blood biochemical parameters were minimally affected. Temporarily holding post-transported Kemp’s ridley turtles in a recovery pool prior to release ameliorated some of these changes [144] and holding in a temporary recovery pool is now encouraged by USFWS after long-duration transport when possible [65]. Similar studies regarding air transport are needed since air transport is being used with increasing frequency. 

The choice of release location for rehabilitated sea turtles may also affect their welfare. The ocean is a dangerous place, and it is impossible to select a release location with no risk. Retrospective and prospective studies are needed on the long-term fate of sea turtles after release from various locations. It is intuitive that areas of high fishing intensity and vessel traffic should be avoided when possible. However, other factors that may affect long-term fate, such as proximity of release sites to breeding areas or seasonal foraging areas, have not yet been studied. 

##### Post-Release Monitoring of Sea Turtles

After release, long-term survival of sea turtles is difficult to determine in the oceanic environment, relying largely on subsequent happenstance. While not an absolute requirement, many rehabilitation facilities acquire USFWS authorization to insert passive integrated transponders (PIT) and/or flipper tags prior to release, following federal specifications for turtle size and tag size [65,74]. However, future detection by these methods requires the physical presence of a human observer and a tag reader (for PIT tags), and flipper tags are known to detach over time. Improving technologies such as satellite and acoustic telemetry provide opportunities for remote detection [74,146]. Since turtles have a life expectancy similar to humans, if not longer, it is difficult to define “success” and welfare after release. Must a rehabilitated sea turtle survive for decades and reproduce after release to be considered “successful”? Relative to the number of sea turtles released from rehabilitation, very few are re-encountered, and even fewer have been observed later nesting [68,74]. Indeed, there has been a call for increased scrutiny of the conservation impact of sea turtle rehabilitation programs [147]. Emerging technology must be used to better understand the long-term fate of a much greater percentage of rehabilitated sea turtles after release [74]. 

#### 4.2.3. Captive Placement of Sea Turtles

While sea turtles are robust, and can often survive with chronic conditions, the wellness of the turtle as well as long-term resource allocation must be considered. A decision toward permanent human care implies a decades-long commitment to meeting the complex environmental, nutritional, and wellness needs of a large marine animal [72]. While the percentages of rehabilitated sea turtles placed under permanent human care are relatively low (0.15% to 1.6%) [69,73], these still represent a substantial number of individuals that require complex, long-term care.

### 4.3. Marine Mammal Welfare

Concurrent growth in human populations along US coastlines, growing, sometimes resurgent, populations of some species of marine mammals, and dynamic and altered ocean, bay and estuary environments are placing humans and marine mammals in closer proximity and presenting risk. This urban: wildlife interface introduces the opportunity for an increased number of negative human interactions with challenging and sometimes fatal results for marine mammals (e.g., female seals abandoning dependent pups in the face of encroaching beach-goers, whales crossing shipping lanes near busy ports leading to fatal strikes, property loss in marinas where pinnipeds haul out, injury to seals, humans or pets from bites when wildlife and managed pets encounter one another). 

Entanglement of whales, dolphins, seals, sea lions and otters occur in the ropes, lines, nets, and other equipment that are part of fisheries, both commercial and recreational. These entanglements can often prove fatal for the marine mammals involved. While quickly fatal entanglements are tragic, sublethal entanglements that leave an animal to tow entangled lines or to endure slowly tightening constrictions of the neck or limbs are a welfare concern of heightened proportion due to the suffering they cause. 

The state of fisheries themselves have bearing on the state of welfare of marine mammals. Consistent and healthy stocks of forage fish are relied on by marine mammals and influence the success of trans-oceanic migrations, the rearing of the young, and more. Fish stocks are impacted by dynamic pressures such as oceanic temperature increase relative to climate change. Notably, a multiple-year occurrence of anomalously warm water in the ocean around California’s Channel Islands impacted the ability of nursing California sea lions to successfully forage for cold-water feed fish while nursing their young. 

Some dependent pups are left unattended on beaches while females forage offshore. The pup is vulnerable to predation, wave action, and more during such periods. Unfortunately, well-meaning members of the public visiting coastal beaches encounter these pups, mistakenly perceive them to be abandoned, and remove them from the beach, effectively orphaning them and requiring their rehabilitation. 

While a marine mammal stranding is an obvious concern for the animal, the situation also poses risks to the public, pets, and property as the occurrence is often on busy and heavily trafficked beaches and waterfronts. A response to a stranded marine mammal is, in fact, a public service performed by responding parties in addition to one driven by animal welfare interests. The risk of disease transmission, injurious bites, and property destruction is very real, and even the act of bringing an animal into care puts a responder at risk. For this reason, intensive classroom and field training is typically conducted for paid staff and volunteer team members of network organizations.

By examining stranded marine mammals, NOAA Fisheries and stranding network partners can better understand causes of mortality and factors that affect marine animal health. Other goals of stranding response efforts are to facilitate humane care out of animal welfare concerns, assist in the recovery of protected species by returning them to the wild whenever possible, and help identify population threats and stressors of marine life, especially stressors that can be prevented or mitigated.

#### 4.3.1. Euthanasia of Marine Mammals

When appropriate, based on a patient’s grave clinical prognosis or at the discretion of the attending veterinarian, euthanasia may be performed. When this is decided, the process must be guided by best practices and specifically by AVMA guidelines for euthanasia [123].

Euthanasia of any marine mammal, whether in a rehabilitation facility or on a beach, warrants extreme care for the welfare of the individual animal, for the professionals attending to the animal, and for those who may be witness to the procedure. For pinnipeds, the procedures are often straightforward, and animals can typically be removed from public locations prior to euthanasia. For small and large cetaceans, the situation can be quite different; the animals are often in highly visible locations and are not easily relocated. The best methods of euthanasia that optimize animal welfare are yet evolving. Because of this, much attention has been placed on both the methodology of euthanasia for cetaceans, especially for large cetaceans, and the consideration of public perception for these procedures [148,149,150,151,152]. 

#### 4.3.2. Release of Marine Mammals to the Wild

To bring a marine mammal into rehabilitation presents its own welfare considerations. The assessment at the stranding location alone can cause stress, as the animal, e.g., a pinniped, is closely observed, cordoned off from other risks of people and pets and otherwise constrained. Capture is inherently stressful and can cause injury, and transportation adds an additional overlay of stress and risk, which can be sustained for an hour or more. Arrival at a rehabilitation facility begins yet another suite of activities, each of which, while necessary to assess, diagnose, treat, or care for the animal, is, by its nature, stressful. 

While in care, marine mammals are at risk of becoming habituated to people to the point that they manifest behavior that is inconsistent with their wild cohorts, putting them at risk for negative human interactions when released. Care environments can be devoid of novel and enriching stimuli for what are often neonate and juvenile individuals. Emerging practices of environmental enrichment and research in enrichment practices can facilitate the development of physical skills and mental and behavioral development in marine mammal patients under care that will better prepare them for release [153,154,155] (Figure 5).

Prior to release, and usually at the time of admission, tags with unique alphanumeric markings are placed on patients, and in certain circumstances, satellite tags are placed on individuals representing endangered species or for those animals of significant interest from a medical or other perspective. Care must be taken in choosing a release site as patients are optimally released in areas that are deemed safe and at which there are conspecifics. 

##### Post-Release Monitoring of Marine Mammals

Following successful rehabilitation, upon clearance from attending veterinarians, and with approval of NOAA Fisheries, marine mammals may be released to the ocean. Such release is conducted under the care of professional staff to ensure the safety of the animal and attending personnel.

Data are limited on animal behavior, activity, and survivorship following rehabilitation. While satellite monitoring of some marine mammal species (e.g., Hawaiian monk seals, bottlenose dolphins, and Steller sea lions) has demonstrated success, there is need for further study. Meanwhile, anecdotal information, such as patient re-sightings with dependent pups, offers positive, if limited, insights [156,157]. 

#### 4.3.3. Captive Placement of Marine Mammals

When a marine mammal patient is deemed to be non-releasable by the attending veterinarian, based on a determination that while otherwise healthy, the patient would likely not be able to thrive in the wild (e.g., in instances such as loss of vision, limb amputation, significant habituation, etc.), captive placement is considered at an accredited zoo or aquarium in a process overseen by NOAA Fisheries. 

In such cases, consideration is paid to the quality of care and habitat provided by the receiving institutions. This includes a consideration of the presence or absence of animals of the same species, gender distribution, personnel training, environmental enrichment capacity, facility accreditation, and more. 

### 4.4. Terrestrial Animal Welfare

Professionals in the field of wildlife rehabilitation may have the opportunity to prevent displacement or unnecessary capture of a wild animal prior to its admission. Many good Samaritans mistake a healthy juvenile for an orphan or misinterpret natural behavior as concerning and interfere with the animal. Additionally, the actions of many humans, such as using certain rodenticides or allowing a pet cat to roam freely outdoors, can be disruptive and even fatal to wildlife. The number of wild mammals killed by domestic cats in the US each year was estimated in 2013 to be between 6.3 billion and 22.3 billion [158]. It is in the best interest of the rehabilitator, the good Samaritan, and, perhaps most importantly, the wild animal, for wildlife rehabilitators to have a strong understanding of the natural history of their region’s wildlife, resources, and techniques for preventing or managing the human–wildlife conflicts, as well as a good rapport with the public to most effectively maintain the quality of life of wild animals. 

#### 4.4.1. Euthanasia of Terrestrial Wildlife

For animals that cannot or should not be returned to the wild, there are two possible outcomes: euthanasia or placement in a permanent captive care situation for the sake of education, fostering, and/or breeding. When euthanasia is deemed the appropriate course of action, it should be performed as soon as possible to delay further suffering and stress. The process should be performed by a veterinarian or by trained personnel following the most current AVMA guidelines [123].

#### 4.4.2. Release of Terrestrial Wildlife to the Wild

The maintenance of an animal’s welfare during the rehabilitation process poses a challenge since simply by being able to be captured, transported, and held in captivity, the animal’s wellbeing declines. Even in cases when the species and nature of injury or illness are familiar and the animal’s physical needs can be appropriately met, maintaining a high-quality mental state of the animal often has limitations. Having an environmental set up that is quiet with a lower level of human-related noises and reduction in sensory stimuli may reduce the stressors (Table 2). Our interpretation of an animal’s quality of life is limited to a few straightforward measurements, including food consumption, display of natural behaviors such as running, climbing, and flying, and its body posture. 

Wildlife rehabilitation requires animals with previously limited or non-existent human contact to be looked at, restrained, and manipulated by human caretakers. In some cases, these interactions include a painful or unpleasant component as is often necessary during a physical exam, wound management and fracture immobilization and diagnostic testing. Despite the necessity of these interactions for the ultimate, hopeful positive outcome of return to the wild, they yield a temporary negative impact on the individual’s welfare that is almost always unavoidable [100]. To combat these unavoidable stressors in the animal, regular use of analgesia for painful conditions, the use of appropriate antimicrobials for infectious diseases, and, in some cases, sedatives or tranquilizers for particularly high-stress species or individuals are critical tools for a rehabilitator. 

The nature of wild animals is often to mask injury or illness, so welfare assessment is further limited by this innate trait. Remote monitoring and telemetry systems improve animal welfare by monitoring stress levels, behavior, assessing response to therapy, or as a diagnostic tool, without needing to restrain the animal [159,160,161,162,163,164,165,166] (Figure 6). Telemetry can also be used to monitor physiologic signs of stress such as an increased heart rate. These physiologic signs of stress can then be correlated with behavioral cues and monitored via non-invasive techniques [124] (Ahlmann-Garcia, unpublished date, 2023).

Enrichment in rehabilitation may also reduce the distress a wild animal experiences in captivity. The goals of enrichment programs in rehabilitation include increased diversity and range of expressed normal behaviors and reduced frequency of abnormal behavior. The benefits of intentional enrichment programs in wildlife rehabilitation can encourage natural behaviors and increase quality of life while an animal is in temporary captivity. There are generally five accepted categories of enrichment. All these categories are customized for each individual depending on the stage in rehabilitation and natural history. The enrichment categories are: social, cognitive, sensory, nutritional, and the physical habitat/enclosure design including cage furniture [167,168]. 

Finally, the use of masks, costumes, and/or puppets during the care of orphaned wild animals may likewise reduce stress associated with overt human contact and reduce the chance of habituation or imprinting (Figure 7) [169,170].

When considering an animal for release, important criteria need to be evaluated before the animal is returned to the wild. First, the animal must be fully recovered; second, the animal must be able to survive in its natural environment, which entails finding food, water, shelter, and mates and defending itself from potential predators. In sum, terrestrial animals should be able to survive in the wild at the same level as their conspecifics [8]. Finally, the animal should have no negative impact on its new environment that would not normally be present from another wild counterpart, as may be the case if a carrier of a rare or novel infectious agent were to be released.

Preparation for release should include reviewing the animal’s original recovery location, preferred natural habitat, seasonal activity, and the effect the newly returned animal will have on the ecosystem to which it is returning. Some species have such a high site fidelity that they should only be returned to their specific point of origin. In fact, in some states, the release of reptile and amphibian species is forbidden if the natural point of origin is unknown. Additionally, due to concern for emerging infectious and highly fatal diseases ravaging many reptiles and amphibians, these animals may have additional restrictions regarding releasing them back to the wild [171]. 

Ironically, the period of transport for release can still have significant negative effects on the welfare of the animal even though the rehabilitation process is effectively over. Transport of wild animals has been documented to increase the risk of infectious disease [172], and a recent review on the effects on animal welfare during transportation of wild mammals found ample evidence of poor welfare during this process including dehydration, transport-induced modulation of the immune system, metabolic changes associated with anorexia, increased heart rate, blood pressure, and body temperature, direct measurements of increased plasma catecholamine and epinephrine levels, and behavioral changes [173]. The review used the Five Domains Model, which uses nutrition, environment, health, behavior, and mental state to assess welfare and reviewed 60 publications. These findings definitively document that while this last phase of the rehabilitation process is brief and minimally invasive, its potential for negative impacts on the animal’s wellbeing are substantial and should not be overlooked (Figure 8).

There are two main types of release options: soft and hard. Age and species usually dictate what type of release is best. Soft releases are selected when the animal needs support for a period of time after it returns to the wild, as may be the case with animals that were originally presented as orphans. The animal is often kept in an enclosure at the release site for a determined amount of time. When ready, the enclosure is opened to allow the animal to leave but it is left in place, often with supplemental food. The animal then has the option to explore its new surroundings while still having some of its physical needs such as food, shelter, and safety provided with the important benefit of minimizing ongoing human contact. A hard release occurs when the animal is released into an appropriate habitat with no ongoing support after release.

##### Post-Release Monitoring of Terrestrial Wildlife

Post-release evaluation of terrestrial animals from rehabilitation settings is often limited but is a growing area of needed research. When performed, this research may use radio telemetry, GPS collars or passive integrated transponders (PIT) tags, among other methods. Results are often highly variable based on the species, reason for admission to rehabilitation and location of release, but a recent study in the *Journal for Nature Conservation* found that by simulating advanced populations in five species of terrestrial animals commonly encountered in rehabilitation settings, rehabilitation efforts have the potential to support recovery and stabilization of populations in some cases [174].

When assessing “success” following the release of a wild animal, one is often focused on the individual rather than the population. Measurements may include long-term survival, integration into the local population if appropriate for the species and successful reproduction of the individual, but success should also focus on the maintained health status of the population and environment into which the animal has been released. A decline in the health of a local population or ecosystem as a result of introducing a rare or novel disease or parasite in the released animal, for example, would likely not be considered successful, despite the survival of the single animal [2].

#### 4.4.3. Captive Placement of Terrestrial Wildlife

Most wild animals presented to wildlife rehabilitation facilities are not suitable for placement. This may be due to permanent and often painful deficits from the severity of the initial sustained injury or illness, the animal’s inability to comfortably perform natural behaviors in captivity, and/or a mental status that will not allow for a good quality of life in permanent captivity. Even for animals that may be candidates for placement, there are often no suitable placement options available. For terrestrial animals, restrictions for placement are usually at the state level.

For placement, the animal should be medically evaluated and should “pass” a behavioral test to gauge its comfort level in captivity. Animals need to be fully healed from any injuries before placement is pursued. Applications from the placing facility and from the receiving facility need to be approved by state regulators, and most states require a letter from the VOR stating why the particular animal is not releasable and would make a good candidate for placement. Facilities placing animals need to be convinced that the receiving facility has the expertise and resources to care for, maintain and train the particular animal. Receiving facilities need to conduct due diligence to be assured that the animal is the “right animal for the job.” The receiving facility should be able to meet the current welfare needs of the animal, including diet and space needs as well as anticipate and be prepared to manage any future problems associated with the original illness or injury that would negatively impact the animal’s wellbeing.

## 5. Conclusions

Hundreds of thousands of wild animals are presented for rehabilitation in the US every year, their lives interrupted. Increasingly, our society recognizes the concept of animal welfare and understands that humans negatively impact the welfare of animals, including wildlife. Many also acknowledge our moral obligation to promote good animal welfare. This paper discusses the intersection between wildlife rehabilitation and wild animal welfare. It presents a comprehensive overview of the practices of wildlife rehabilitation in the US for multiple taxa, and the complexity of considerations rehabilitators and veterinarians face while trying to optimize the welfare of wildlife in need of care and rehabilitation.

The goal of wildlife rehabilitation is to release an animal back into the wild, a resumption of its life’s path. However, this oft-stated goal ignores reality. In fact, we have no idea what this “second chance” holds for all but a few of the animals that are released. This goal also neglects the reality of the other outcomes of rehabilitation—euthanasia, death in care, and captive placement. Maintaining the individual animal’s welfare at the center of the rehabilitation process requires deliberate, timely and humane decision making. Wildlife rehabilitation is costly in terms of animal welfare, financial resources, and people’s time and wellness. Improving the welfare of wild animals before, during, and after rehabilitation requires strengthening and harmonizing regulatory policies, investment in evidence-based science in rehabilitation, clinical wildlife medicine, and post-release monitoring, as well as forward-thinking and inclusive ethics, including the following items.

Preventing Anthropogenic Causes of Admission. Increasing numbers of animals are being admitted due to anthropogenic causes related to human activities [3,116]. Examining the reasons of wildlife admission provides an opportunity to work upstream to develop mitigation strategies. Preventing the need for rehabilitative care is the best way to improve the welfare of wildlife.Providing Resources and Support for Rehabilitation. In the US, wildlife rehabilitation is largely paid for by private individuals and donations. The adoption of established best practices may be limited by physical and/or financial resources. Having public funding at the federal, state, or county level specifically directed to wildlife rehabilitation would allow for improved care and outcomes of wild animals. Although urban wildlife management is a growing field within state fish and game departments, a dedicated revenue stream for wildlife rehabilitation has yet to be identified [171,175].Continued Development of Evidence-based Wildlife Rehabilitation Methods and Welfare Measures. The advent of electronic database programs specific to wildlife rehabilitation [176,177,178,179] has resulted in greater access to large data sets. These data provide vital information for making better care choices for individuals and facilities. By combining these data in collaboration with other institutions and data from post-release monitoring, wildlife rehabilitation professionals can develop protocols with the highest likelihood of successful outcomes, and concurrently improve the welfare of all animals under care [2,25,31,118,174,180].Greater Engagement of the Veterinary Profession. Wildlife welfare would benefit greatly from increased veterinary involvement in wildlife rehabilitation. All animals need a rapid and thorough assessment within the first few hours of presentation. Only veterinarians can provide diagnostic testing, surgical care, access to analgesics, the most humane euthanasia methods, and recommend captive placement. There is training available for veterinarians and veterinary students [181,182].Harmonization of Regulatory Oversight with Standards of Care, Training, and Accountability. Federal and state regulatory policies should be reviewed and updated on a regular schedule. These reviews should incorporate the most recent evidence-based science and enhance standards of rehabilitative care, training, and accountability to ensure optimal welfare for wildlife. Increasing harmonization of wildlife rehabilitation policies within federal programs, between state programs, and between federal and state programs would also improve wildlife welfare by ensuring prompt care and allowing for more data aggregation. Ongoing collaborations between federal and state level regulators, wildlife rehabilitators, and clinical wildlife veterinarians help prepare for and mitigate the challenges ahead for wildlife and rehabilitation.Increasing Public Education. Interactions between urban wildlife and the public are increasing. Stimulating interest in local wildlife to prevent conflict, countering misinformation about animal behavior, and providing tools for successful coexistence with wildlife is another strategy to reduce the number of animals in need of rehabilitative care [30,183,184].

## Figures and Tables

**Figure 1 animals-13-01836-f001:**
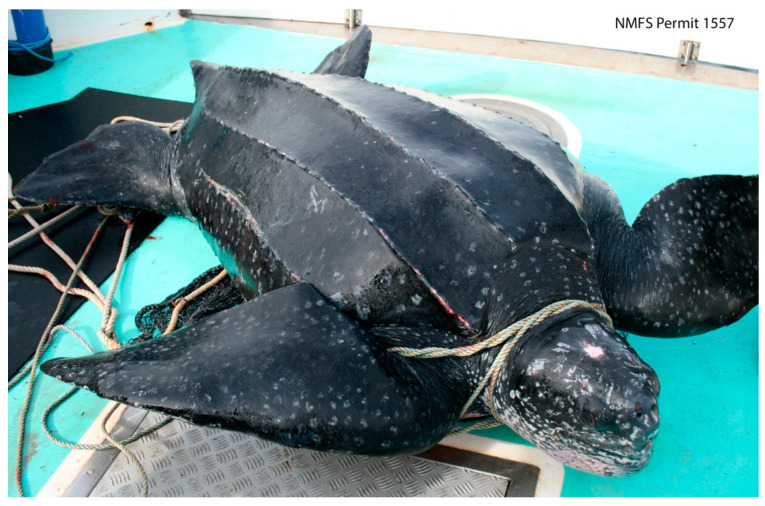
Leatherback sea turtle (*Dermochelys coriacea*) entangled in fishing gear. Many turtles are entangled in active gear, involving the turtles’ necks and/or front flippers. Photo courtesy of New England Aquarium.

**Figure 2 animals-13-01836-f002:**
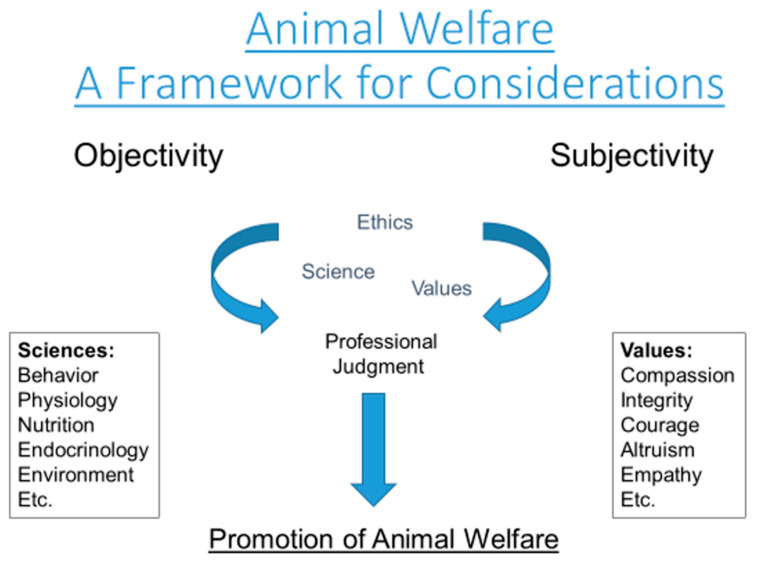
Animal welfare framework for wildlife rehabilitation. Animal welfare is optimized by utilizing objective policies and science, and subjective ethics and values. Graphic created by Jeff Boehm; Microsoft 365.

**Figure 3 animals-13-01836-f003:**
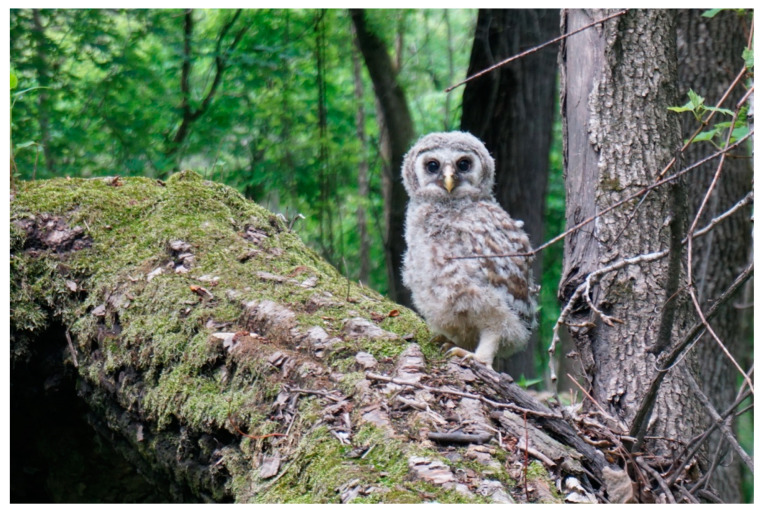
Renesting juvenile barred owl (*Strix varia*). Every effort should be made to renest juvenile birds in natal or foster nests. Photo courtesy of The Raptor Center.

**Figure 4 animals-13-01836-f004:**
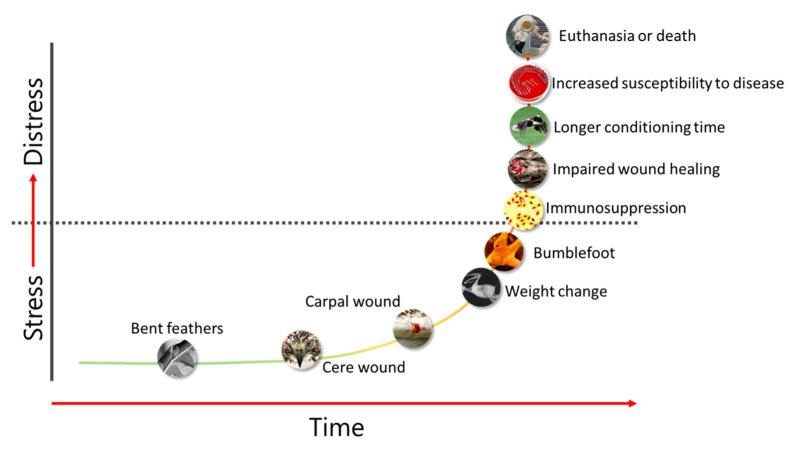
Time is trauma. Rehabilitators and veterinarians need to constantly re-evaluate an animal’s prognosis for release to the wild. Graphic created by Michelle Willette; Microsoft 365.

**Figure 5 animals-13-01836-f005:**
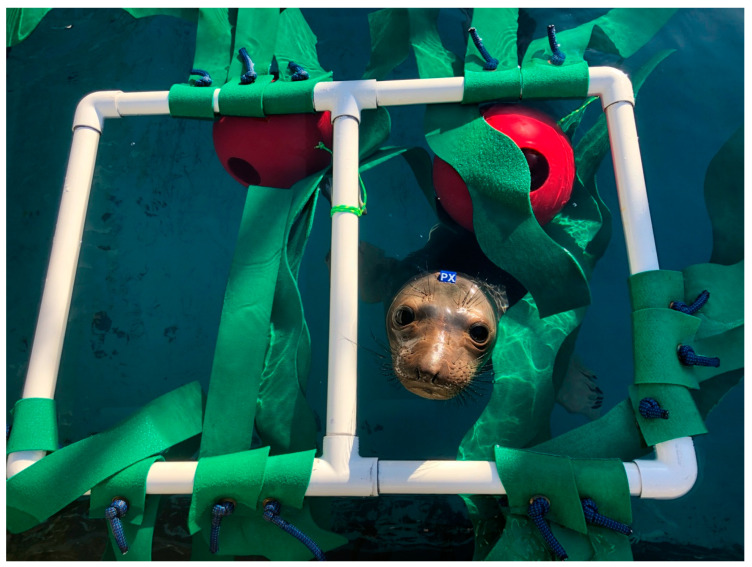
A Northern elephant seal (*Mirounga angustirostris*) interacts with an artificial kelp device. Food can be hidden in this device, which then sinks to encourage diving and breath holding. A biodegradable tag affixed with surgical glue is pictured here on the head of the patient to facilitate patient identification while under care. Photo courtesy of the Marine Mammal Center.

**Figure 6 animals-13-01836-f006:**
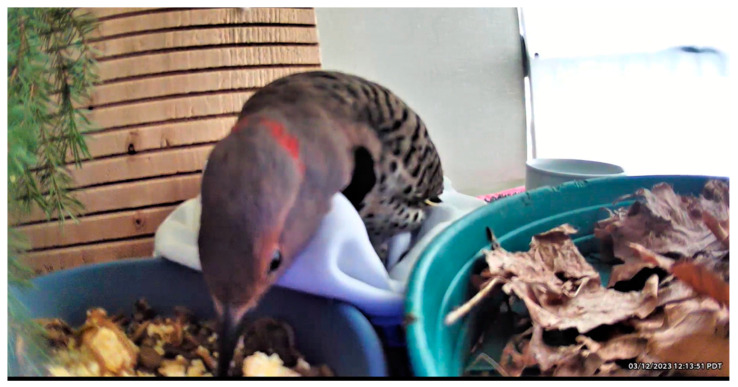
Remote monitoring of a Northern flicker (*Colaptes auratus*). Remote cameras allow for the monitoring of an animal’s physical, emotional, and behavioral health without disturbance. Photo courtesy of the Progressive Animal Welfare Society.

**Figure 7 animals-13-01836-f007:**
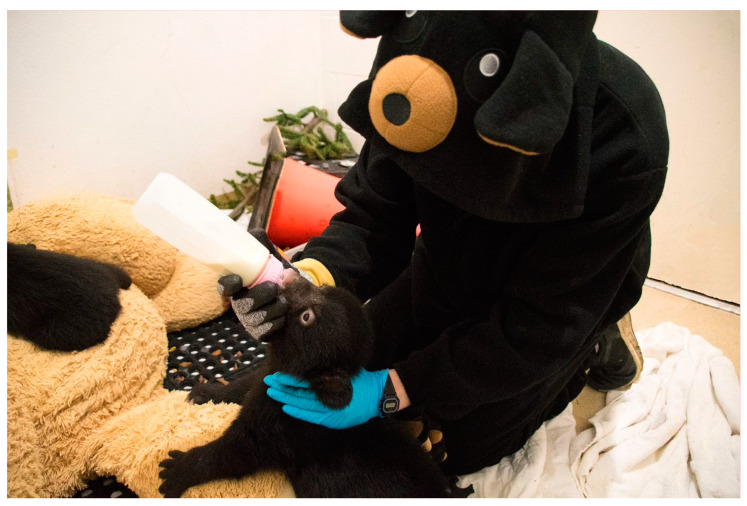
Use of costume while feeding an orphaned black bear cub (*Ursus americanus*). Using disguises may reduce stress and habituation or imprinting due to human contact. Photo courtesy of the Progressive Animal Welfare Society.

**Figure 8 animals-13-01836-f008:**
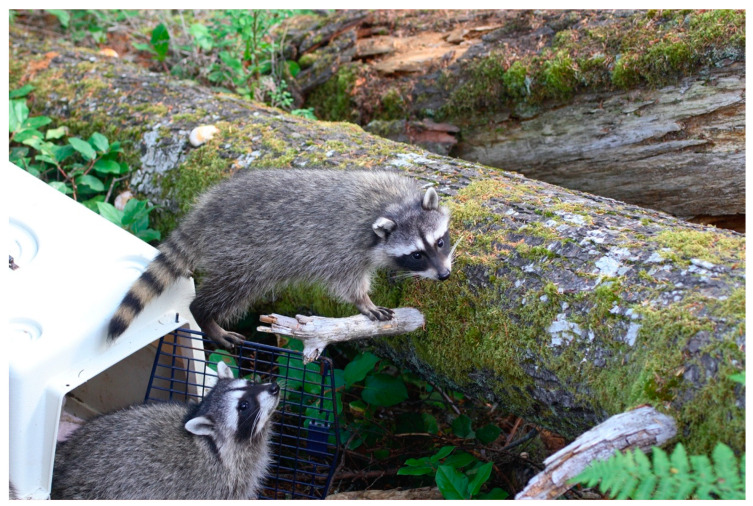
Post-release juvenile raccoons (*Procyon lotor*) calmly exploring the environment. While the release process is brief, its potential for negative welfare impacts is substantial. Photo courtesy of the Progressive Animal Welfare Society.

**Table 1 animals-13-01836-t001:** Outcome criteria for wildlife rehabilitation. These state and species-/taxonomic-specific charts should incorporate the mission of the organization, any regulatory restrictions, physical, mental, and behavioral assessments, and the availability of resources required for the bird. Table courtesy of Partners for Wildlife.

Criteria	Euthanasia	Release	Captive Placement
Mission			Rare; if release fails, assess physical, emotional, behavioral, and environmental aspects for placement
Legal Restrictions			
Federal	Appropriate federal MBTA Permit—Rehabilitation MBTA—any bird that cannot feed itself, perch upright, or ambulate without inflicting additional injuries to itself where medical and/or rehabilitative care will not reverse such conditions; any bird that is completely blind; and any bird that has sustained injuries that would require amputation of a leg, a foot, or a wing at the elbow or above (humeroulnar joint) MBTA—obtain authorization before euthanizing endangered and threatened migratory bird species… without USFWS authorization when prompt euthanasia is warranted by humane consideration for the welfare of the bird.	Appropriate federal MBTA Permit—Rehabilitation	MBTA Permit—Veterinarian Statement of Condition that Renders Migratory Bird/Eagle as Non-Releasable MBTA Permit—Special Purpose—Possession for Education (Live) BGEPA Permit—Eagle Exhibition
	Drug Enforcement Agency—Controlled Substances Act	Drug Enforcement Agency—Controlled Substances Act Food and Drug Agency— Federal Food and Drug Cosmetic Act (AMDUCA)	
State	Appropriate state wildlife rehabilitation permit Veterinary Practice Acts Pharmacy Schedule Reportable Diseases List	Appropriate state wildlife rehabilitation permit Veterinary Practice Acts Pharmacy Schedule Reportable Diseases List	Appropriate state wildlife placement permit
Assessments			
Physical/Body	Non-repairable bony fractures or soft-tissue injuries Malunion or healed malaligned fracture Joint trauma including fractures/luxations/subluxations Partial wing amputations * >10% decrease in long bone length (except femur) Significant trauma to patagium and/or patagial tendon * Coelomic trauma with exposed/contaminated internal organs “Spinal trauma” with posterior paralysis and/or no “deep pain” response Severe toxicity with clinical signs * Severe starvation * Permanent eye injury (injuries) with vision deficits * Severe pododermatitis (Grade 4/5) Amputation of 1st and/or 2nd digit * “Three Strikes Rule” *	Full recovery from illness and/or injuries with no clinical evidence of complications Goals of physical reconditioning program met with normal flight mechanics, strength, and endurance for species A complete set of healthy remiges and rectrices (natural or imped) No risk to human/domestic animal/wildlife/environmental health and safety	Injuries are completely healed Illness or injury will not progress Permanent injury compatible with captive purpose
Emotional/Mind			Prior injury/illness is not painful Adapting well to captivity
Behavioral/Nature	Exhibits inappropriate behavior due to medical condition Hard “malimprint” or habituated *	Exhibits species-appropriate natural behaviors Able to acquire live prey	Exhibits species-appropriate behavior Injury/illness will not prevent opportunities to thrive (perch, move, balance, self-feed, bathe, and conduct species appropriate behaviors) Does not exhibit self-destructive or stereotypic behaviors Not aggressive towards humans
Resources (Required vs Available)	Training as appropriate for method Licensing as appropriate for method (chemical euthanasia) Availability of proposed drug for chemical euthanasia	Resources required for care and/or treatment, including facility/equipment, diet, knowledge, skills Understanding of species biology, natural history, welfare needs and behavior Rehabilitation of this individual will not adversely affect the facility’s operation Availability of nest/foster nest Appropriate release protocol Identification and post-release monitoring if possible	Appropriate placement available, including permits, resources, knowledge, training and welfare standards Appropriate acquisition for facility’s collection plan Access to veterinary services Appropriate housing available Environment appropriate to exhibit natural behaviors
Professional Standards Best Practices References	AVMA Guidelines for Euthanasia of Animals (Current Edition) NWRA Wildlife Formulary (Current Edition) AVMA Professional Code of Ethics NWRA Professional Code of Ethics	NWRA Minimum Standards for Wildlife Rehabilitation (Current Edition) NWRA Principles of Wildlife Rehabilitation (Current Edition) IWRC Wildlife Rehabilitation—A Comprehensive Approach (Current Edition)	Raptors in Captivity: Guidelines for Care and Management (Current Edition) AZA Institutional Ambassador Animal Policy/Placement Tool Wildlife In Education: A Guide for the Care and Use of Program Animals (Current Edition) GFAS Standards for Birds of Prey Sanctuaries (Current Edition) NWRA Professional Code of Ethics—Educators

* Evaluated on a case-by-case basis.

**Table 2 animals-13-01836-t002:** Mitigating captive stress in wildlife. This table presents common stress factors for wildlife in rehabilitation with suggested practice and administrative controls. Reprinted with permission from Miller and Schlieps, Standards for Wildlife Rehabilitation; National Wildlife Rehabilitators Association, 2021.

Means to Reduce Stress—Practice Controls	Means to Reduce Stress—Administrative Controls
Proximity to humans	
Use peripheral vision/avoid direct eye contact Provide visual barriers Provide species-appropriate ambient sound Provide hides/retreat spaces Avoid placing enclosures in high-traffic areas Reduce time in captivity when it no longer benefits the patient Set rules for media interaction	Foster a work culture that is focused on stress reduction and patient advocacy Be mindful of the experience the patient is having at all times Educate all staff, volunteers, and visitors about stress and stress-reduction practices Develop written policies about talking, answering phones, or otherwise attempting to multitask while caring for patients Develop written media guidelines for rescue or release Have policies in place regarding appropriate behavior around patients
Restricted movement/Reduced retreat space	
Offer complex enriched environments; Offer opportunities for exploration Mimic natural surroundings and substrates Provide hides/retreat spaces if space allows Provide visual barriers on the enclosure in lieu of retreat space Reduce captive auditory stressors	Develop an enrichment plan that includes elements related to all senses Develop a matrix that identifies enrichment opportunities for patients at different activity levels or rehabilitation phases
Foreign environment and loss of control/decision-making capacity	
Use natural history to inform decisions; observe seasonal photoperiod and humidity Use appropriate indoor lighting; avoid use of fluorescent lights Mimic normal thermal environments when possible; mimic natural surroundings and substrates Offer complex enriched environments Provide opportunities for choice (e.g., food items, enrichment items, shelter options)	Assign one person (or a team of volunteers) to create enrichment opportunities for in-house patients on a weekly basis Develop an enrichment calendar so that types of enrichment offered are rotated on a regular basis Design larger enclosures with areas of different temperatures, humidity, light, hides, or other options, to diversify choices
Visual, auditory, and olfactory stressors	
Avoid direct eye contact and sudden movements Refrain from wearing jewelry, bold patterns, or bright colors Be mindful of light intensity and light selection; avoid use of fluorescent lights Use cage covers that let light through while still providing visual barrier when appropriate Be aware of sounds from routine tasks, and try to mitigate them Use white-noise machines or appropriate nature recordings to mask unnatural sounds (e.g., use recordings of ocean sounds for seabirds and shorebirds) Use natural colors within the facility (e.g., walls, floors, artwork, uniforms, sheets/towels) Use unscented cleaning products whenever possible; refrain from using perfumes or scented lotions Do not smoke or use car air fresheners in transport vehicles Avoid placing enclosures in high-traffic areas	Create a culture where consideration of patient’s audio experience is prioritized Develop written dress code policy Develop written policies to reduce unnecessary “peeking” into enclosures by staff and volunteers Develop written policies to reduce speaking in clinic or rehabilitation center, or with patients in hand Model behavior for others Design interior spaces to take advantage of natural light sources
Capture, handling, and restraint	
Use passive capture techniques that do not require handling, and limit chase Use experienced personnel for difficult captures Allow new patients to rest and recover from capture, handling, and transport stress prior to performing exams unless immediate life-saving treatment is required Use firm but gentle restraint Keep patient’s head covered unless it is being examined Have tools and supplies necessary for treatment to reduce time in hand Have a handler for exams/treatments when appropriate Limit readjustments and movement of the animal on the exam table (e.g., the examiner can move when possible rather than move the animal) Consider natural history and physical build of the patient to identify ways to increase comfort (e.g., memory foam can be added to exam tables for large-bodied birds to relieve pressure on the keel; mirrors can be used to examine the undersides/bellies of porcupines) Reduce need for capture and handling in the facility by use of remote monitoring Evaluate the necessity of handling and weigh the cost to the benefit of treatment Use retractable curtains, temporary walls, or other physical barriers to reduce enclosure dimensions during capture	Have written transport policies and protocols for volunteers and staff Design and arrange enclosure furnishings to ease capture Design new enclosures to ease capture (e.g., built-in capture chutes, dens with doors that can be closed remotely, hospital cages with dividers) Know the goals for each exam or treatment
Pain	
Provide appropriate substrates and padding as needed; medically manage pain Euthanize non-treatable animals as soon as possible to prevent further suffering	Stay current on pain management protocols and support therapies Develop written euthanasia guidelines and protocols specific to the species in rehabilitation and the resources available at the facility Network and consult with other rehabilitators on difficult decisions, particularly rehabilitators with species’ expertise
Proximity to nonhuman predators	
Provide visual and auditory barriers House predator and prey species in separate areas House predators out of visual and hearing range of prey species	Map out enclosures with species designations Consider predator/prey proximity when designing enclosures
Disturbance (peeking, observations, cage cleaning, feeding)	
Establish routines; bundle tasks; reduce disturbance Prohibit or restrict visitors Move slowly; avoid direct eye contact Use remote cameras, baby monitors, or ceiling-mounted mirrors to observe behavior without handling	Have a written policy regarding visitors and public access Patients should not be able to see members of the public at any time (see USFWS 50 CFR 21.71)
Maintenance in abnormal social groups	
House with conspecifics Provide foster parent when appropriate; reduce the number of individuals per enclosure Do not transport or house wildlife patients with domestic animals	Develop a policy for transferring single juvenile animals to a facility with conspecifics (and form a network to facilitate transfers) Have a written policy for a number of animals to be housed in each enclosure Have a written policy establishing limits on intake of number of individuals per species
Reduced feeding opportunities	
Offer complex feeding challenges (e.g., food hidden in environment, food inside objects) Use native plants when possible Use techniques to provide natural live foods, such as fruit bins to attract edible insects (e.g., fruit flies) Place food in a location where animals feel comfortable feeding (e.g., away from entrance)	Develop an enrichment plan that includes elements related to all senses Develop a matrix that identifies enrichment opportunities for patients at different activity levels or rehabilitation phases
Diet change	
Use native plants and seasonal/natural food sources when appropriate Use natural history to inform methods of food presentation Identify tricks for encouraging difficult eaters in captivity; leave enough food of multiple varieties in enclosures	Network with species specialists about best foods, tricks and techniques Calculate metabolic requirements and deficits to determine the amount to be fed Develop a network to gather natural foods (e.g., local fishermen, gardeners, naturalists to collect insects)

## Data Availability

Not applicable.

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
