# Peer review of "Interrupted Lives: Welfare Considerations in Wildlife Rehabilitation"

_animals, 2023, doi:10.3390/ani13111836_

Round 1

Reviewer 1 Report

Summary: This review provides a broad review of all the many challenges that wildlife in rehabilitation settings face in the United States. A unique feature of this paper includes the focus on both terrestrial and aquatic wildlife rehabilitation settings.

Minor Comments:

Line 53: Introducing the concept of welfare with an anthropogenic disturbance focus ("express their concerns about how our actions affect the quality of life of animals") may not be an appropriately broad introduction of the concept of welfare.  The welfare of an animal can be assessed in scenarios where human actions are not directly affecting the quality of life of an animal, as well.

Sections 2, 3, & 4: I appreciate the authors desire to provide context for wildlife rehabilitation regulations and standards in various taxa. I suggest that the authors combine the sections and arrange them by taxa (e.g. sea turtles regulations, standards of care, and specific welfare issues) because this may help readers better understand and use the sections for their taxa of interest and allow for improved brevity throughout the paper.

Line 503: These introductory paragraphs of Section 4 may be better integrated into the introduction (Section 1), so as to not be redundant.

Line 510: It may also be prudent to include the American College of Animal Welfare in this list of the definitions of welfare available.

Line 559: Changing the title as "Avian Welfare" and specifically highlighting migratory birds throughout may be a more accurate solution for this section.

Author Response

See attached.

Thank you for your time on our behalf.

The authors.

Reviewer 2 Report

See Attached file.

This is one of the better written manuscripts that I have read of late.  Most of the issues with language relate to the formatting of the references, all of which can be easily corrected.
